# Inhibition of KRAS, MEK and PI3K Demonstrate Synergistic Anti-Tumor Effects in Pancreatic Ductal Adenocarcinoma Cell Lines

**DOI:** 10.3390/cancers14184467

**Published:** 2022-09-14

**Authors:** Yixuan Ma, Benjamin Schulz, Nares Trakooljul, Moosheer Al Ammar, Anett Sekora, Sina Sender, Frieder Hadlich, Dietmar Zechner, Frank Ulrich Weiss, Markus M. Lerch, Robert Jaster, Christian Junghanss, Hugo Murua Escobar

**Affiliations:** 1Department of Medicine Clinic III, Hematology, Oncology and Palliative Medicine, Rostock University Medical Center, 18057 Rostock, Germany; yixuan.ma@med.uni-rostock.de (Y.M.); moosheer.alammar@med.uni-rostock.de (M.A.A.); anett.sekora@med.uni-rostock.de (A.S.); sina.sender@med.uni-rostock.de (S.S.); christian.junghanss@med.uni-rostock.de (C.J.); 2Institute for Experimental Surgery, Rostock University Medical Center, 18057 Rostock, Germany; benjamin.schulz@med.uni-rostock.de (B.S.); dietmar.zechner@uni-rostock.de (D.Z.); 3Institute of Genome Biology, Research Institute for Farm Animal Biology (FBN), 18196 Dummerstorf, Germany; trakooljul@fbn-dummerstorf.de (N.T.); hadlich@fbn-dummerstorf.de (F.H.); 4Department of Medicine A, University Medicine Greifswald, 17475 Greifswald, Germany; ulrich.weiss@med.uni-greifswald.de (F.U.W.); markus.lerch@med.uni-muenchen.de (M.M.L.); 5Ludwig Maximilian University Hospital, Ludwig Maximilian University of Munich, 81377 Munich, Germany; 6Department of Medicine II, Division of Gastroenterology, Rostock University Medical Center, 18057 Rostock, Germany; robert.jaster@med.uni-rostock.de

**Keywords:** pancreatic ductal adenocarcinoma, KRAS, kinase inhibitors, gene expression

## Abstract

**Simple Summary:**

Small molecule inhibitors and targeted therapy are considered to have significant potential for pancreatic ductal adenocarcinoma therapies. Preclinical studies of novel inhibitors and inhibitor combinations can elucidate their acting mechanisms and provide valuable data for in vivo research and clinical trials. We explored the antitumor efficacy of KRAS inhibitors BI-3406 and sotorasib alone or in combination with the downstream inhibitors trametinib and buparlisib in PDAC cell lines, characterized by different *KRAS* mutational statuses. The two KRAS inhibitors demonstrated different anti-tumor efficacy and displayed synergistic or additive effects, when combined with downstream pathway inhibitors. These data emphasized the importance of KRAS as a therapeutic target for PDAC and indicate two distinct mechanisms of KRAS inhibition and their interactions with downstream pathway inhibitors.

**Abstract:**

Kirsten rat sarcoma virus (*KRAS*) mutations are widespread in pancreatic ductal adenocarcinoma (PDAC) and contribute significantly to tumor initiation, progression, tumor relapse/resistance, and prognosis of patients. Although inhibitors against *KRAS* mutations have been developed, this therapeutic approach is not routinely used in PDAC patients. We investigated the anti-tumor efficacy of two KRAS inhibitors BI-3406 (KRAS::SOS1 inhibitor) and sotorasib (KRAS G12C inhibitor) alone or in combination with MEK1/2 inhibitor trametinib and/or PI3K inhibitor buparlisib in seven PDAC cell lines. Whole transcriptomic analysis of combined inhibition and control groups were comparatively analyzed to explore the corresponding mechanisms of inhibitor combination. Both KRAS inhibitors and corresponding combinations exhibited cytotoxicity against specific PDAC cell lines. BI-3406 enhance the efficacy of trametinib and buparlisib in BXPC-3, ASPC-1 and MIA PACA-2, but not in CAPAN-1, while sotorasib enhances the efficacy of trametinib and buparlisib only in MIA PACA-2. The whole transcriptomic analysis demonstrates that the two triple-inhibitor combinations exert antitumor effects by affecting related cell functions, such as affecting the immune system, cell adhesion, cell migration, and cytokine binding. As well as directly involved in RAF/MEK/ERK pathway and PI3K/AKT pathway affect cell survival. Our current study confirmed inhibition of KRAS and its downstream pathways as a potential novel therapy for PDAC and provides fundamental data for in vivo evaluations.

## 1. Introduction

Kirsten rat sarcoma viral oncogene homolog (*KRAS*) is one of the most frequently mutated oncogenes in human pancreatic ductal adenocarcinoma (PDAC); oncogenic *KRAS* mutations can be detected in approximately 92% of the PDAC genomes [1,2,3,4,5,6]. The *KRAS* gene encodes the protein KRAS, which is a guanosine triphosphatase (GTPase), and regulates signal transduction by cycling between active guanosine triphosphate (GTP) bound and inactive guanosine diphosphate (GDP) bound statuses [7]. *KRAS* point mutations downregulate the GTPase activity of RAS and prevent the GTPase from promoting the conversion of GTP to GDP. The status of permanent GTP-binding activates downstream signaling pathways, such as the PI3K/AKT pathway or RAF/MEK/ERK pathway, which in turn leads to the initiation and development of PDAC [8]. Moreover, *KRAS* cooperates with other common oncogenes, such as *TP53*, *CDKN2A*, *BRCA3*, *SMAD4*, etc., to cause the initiation and development of PDAC [9,10,11,12,13].

*KRAS* mutations not only cause the initiation and development of PDAC, but they also affect the efficacy of treatment routines and the long-term survival of patients. A considerable number of studies have revealed that *KRAS* mutations lead to a poor prognosis for patients, regardless of whether they undergo surgery [14]. At the same time, a study pointed out that *KRAS* activation plays an important role in the resistance to gemcitabine treatment and relapse after treatment [15]. Another study reported that specific *KRAS* mutation subtypes (G12V, G12D, and G12A) shortened the median overall survival of PDAC patients [16].

Due to the important role of *KRAS* in PDAC, a growing number of studies consider *KRAS* as a target for the treatment of PDAC. Sotorasib is the first small molecule inhibitor against KRAS G12C mutations and was approved by the FDA for the treatment of non-small cell lung cancer (NSCLC) in 2021. Studies have reported that it can effectively inhibit various cell lines that carry *KRAS* G12C mutations, including PDAC cell lines [17]. According to the recently disclosed CodeBreaK 100 clinical trial results, sotorasib displayed good efficacy in the treatment of advanced KRAS G12C-mutated PDAC, with 8 of the 38 patients having a partial response and 32 of 38 patients displaying disease control. The side effects of sotorasib are described as mild, as only a few patients were affected by grade 3 diarrhea, fatigue, and abdominal pain; no grade 4 side effects were observed in the patients [18]. Currently (2022), there are 18 clinical trials targeting KRAS by sotorasib in progress [19]. However, almost all the clinical trials target NSCLC and colorectal cancer and only a very small number of PDAC patients are enrolled. In addition, other reported KRAS G12C inhibitors (adagrasib, JNJ-74699157, and LY3499446) have also achieved distinct effects in cell experiments, and corresponding clinical trials are also ongoing [20,21]. At the same time, inhibitors that directly target other KRAS mutations (e.g., KS-58 targeting KRAS G12D) are under development.

Although the KRAS G12C inhibitors achieved satisfactory effects on its corresponding mutation, *KRAS* G12C mutations accounted for only 1.42% of all *KRAS* mutated PDAC patients. The specific inhibitors for *KRAS* G12D and G12V mutations, which currently represent the majority (40.45% and 32.14%, respectively), are still under development and have not yet entered any clinical trials [22]. Therefore, how to target other types of *KRAS* mutations is also an urgent problem to be solved. It is well known that there are dynamic positive feedback and negative feedback regulation loops in the RAS signaling pathway. A key role in this feedback regulation is the guanine exchange factor son of sevenless 1 (SOS1) [23]. In unstimulated cells, SOS1 hyperphosphorylation caused by mitogen-activated protein kinase (MAPK) activation catalyzes the activation of RAS. At the same time, SOS1 hyperphosphorylation in stimulated cells will cause it to separate from cytosolic glutathione reductase (GRC2) and cause RAS inactivation [23,24]. Moreover, down-regulation or loss of SOS1 lead to a decrease in the survival rate of tumor cells carrying *KRAS* mutations [25]. Based on these studies, Hoffman et al. developed an inhibitor BI-3406 that can block the interaction between SOS1 and KRAS. It can effectively inhibit a variety of *KRAS* mutant tumor cell lines in vivo and in vitro, including *KRAS* G12C/V/S/A, and G13D, and also achieved excellent efficacy in the PDAC cell line MIA PACA-2. Moreover, the experimental animals displayed good tolerance to BI-3406 treatment [26]. Therefore, BI 1701963, another inhibitor closely related to BI-3406, has entered phase I clinical trials.

Although studies on the inhibition of *KRAS* have achieved encouraging results, there are still limitations that exist, especially for PDAC. At present, most studies still focus on NSCLS, while little attention has been paid to PDAC. There are also few studies that investigate the combined application of multiple inhibitors. In the existing studies on PDAC, only the MIA PACA-2 cell line was investigated. As a result, we were unable to evaluate the efficacy of these *KRAS* inhibitors on PDAC cells carrying other *KRAS* mutations. Therefore, we studied the efficacy of multiple *KRAS* mutation inhibitor BI-3406 and specific *KRAS* mutation inhibitor sotorasib in different *KRAS* mutations and wild-type *KRAS* PDAC cell lines. At the same time, we explored the efficacy of *KRAS* inhibitors and their downstream pathways (PI3K/AKT/mTOR pathway and RAF/MEK/ERK pathway) inhibitors in combination. RNA sequencing was performed after the combined application to explore the mechanism of the influence of the multi-inhibitor combination on the pathway.

## 2. Materials and Methods

### 2.1. Kinase Inhibitors

BI-3406 (KRAS::SOS1 inhibitor) was purchased from Chemietek (Chemietek, Indianapolis, IN, USA), sotorasib (KRAS G12C inhibitor), buparlisib (pan-PI3K inhibitor), and trametinib (MEK1/2 inhibitor) were purchased from Selleck Chemicals (Absource Diagnostics GmbH, Munich, Germany). According to the manufacturer’s instructions, all inhibitors were separately dissolved in dimethyl sulfoxide (DMSO) (Sigma Aldrich Chemie GmbH, Steinheim, Germany) as a stock solution, at a final concentration of 10 mM. The stock solutions were stored at −80 °C and diluted into corresponding working concentrations before each experiment.

### 2.2. Cell Lines and Cell Culture

PDAC cell lines ASPC-1, BXPC-3, CAPAN-1, COLO357, PATU8902, and T3M4 were kindly provided by the University Medicine Greifswald and MIA PACA-2 was kindly provided by Prof. Robert Jaster from Rostock University Medical Center. ASPC-1, BXPC-3, COLO357, and T3M4 were cultured in RPMI1640 medium (PAN-Biotech, Aidenbach, Germany), supplemented with 10% heat-inactivated fetal calf serum (FCS) (PAN-Biotech) and 1% penicillin-streptomycin solution (P/S) (10,000 U/mL penicillin, 10 mg/mL streptomycin) (PAN-Biotech). CAPAN-1 was cultured in RPMI1640 medium, supplemented with 15% heat-inactivated FCS and 1% P/S solution. MIA PACA-2 was cultured in DMEM medium (PAN-Biotech), supplemented with 1% heated-inactivated FCS and 1% P/S solution. PATU8902 was cultured in DMEM/F12 medium (PAN-Biotech), supplemented with 10% heated-inactivated FCS and 1% P/S solution. After verifying that all cell lines were not contaminated by mycoplasma, these PDAC cell lines were maintained in a 5% CO_2_ incubator with a humidified atmosphere at 37 °C.

### 2.3. Inhibitor Application Experiments

For the single inhibitor application experiments, the PDAC cell lines were seeded at a density of 3.3 × 10^4^ cells per milliliter in a 24-well plate (in total, 1.5 mL per well, for cell proliferation assay) or a 96-well plate (in total, 150 μL per well, for biomass quantification assay). After 24 h, the supernatant was discarded and media containing increasing concentrations (range from 0.1 to 10 μM for BI-3406, 0.001 to 10 μM for sotorasib) of inhibitors or vehicle (DMSO, as control) were added to the corresponding PDAC cell lines.

The results of single inhibitor application and related experiments were comprehensively analyzed, and specific PDAC cell lines and inhibitor concentrations were selected for further combined application experiments and the concentrations are listed in Table 1 (the results of the buparlisib inhibition assay are detailed in a previously published paper, and the results of the trametinib inhibition assay are detailed in the Appendix A) [27]. Inhibitor concentrations are displayed in Table 1. The PDAC cell lines were seeded in 6-well plates (for RNA isolation), 24-well plates (for proliferation assay, morphological examination, and apoptosis/necrosis analysis), or 96-well plates (for biomass quantification assay). After 24 h, the supernatant was discarded and media containing different combinations of inhibitors were added to the corresponding PDAC cell lines.

The treated cells were incubated for 72 h at 37 °C with 5% CO_2_. At the indicated time points, all cell experiments evaluated at least three biologically independent replicates.

### 2.4. Cell Viability Assays

#### 2.4.1. Proliferation

Proliferation was evaluated by absolute counting, which was determined by trypan blue (Sigma-Aldrich Chemie GmbH, Steinheim, Germany) staining. After inhibitor exposure in 24-well plates, the cells were harvested and washed with 1× PBS (PAN-Biotech). Following the cells being stained with trypan blue, the number of viable cells was determined by counting with a hemocytometer. Proliferation was expressed as a percentage of viable cells treated with the inhibitor to the vehicle-treated control (control = 100%).

#### 2.4.2. Biomass Quantification

Biomass quantification was carried out by crystal violet (CV) staining. After exposure to the corresponding inhibitors, cells in 96-well plates were washed once with PBS and stained with 50 μL 0.2% CV solution on a shaker at room temperature for 10 min. Thereafter, the plates were washed twice with PBS. To elute bound CV, 100 μL 1% sodium dodecyl sulfate (SDS) was added to each well and incubated on a shaker at room temperature for 10 min. Finally, absorbances at a measuring wavelength of 570 nm and at the reference wavelength of 620 nm were measured by a Promega GloMax^®^-Multi Microplate Multimode Reader. The absorbance value of the reference wavelength was subtracted from that of the corresponding measuring wavelength. The value of cells exposed to the vehicle was used as a control and the value of culture media was used as the background. The background value was subtracted from the control and experimental values. The amount of CV directly correlates to cell biomass. The result is expressed as a percentage of the inhibitor-treated group to vehicle-treated controls (control = 100%).

### 2.5. Apoptosis and Necrosis Analyses

Apoptosis and necrosis were evaluated by Annexin V FITC (Becton, Dickinson and Company, Heidelberg, Germany) and propidium iodide (PI) (Sigma-Aldrich Chemie GmbH) double staining by flow cytometry. After exposure to the vehicle control, and both single and combined inhibitors, cells were harvested and washed twice with cold PBS. After the washing step, the cell pellet was resuspended in 100 µL Annexin V binding buffer (1×) (Becton, Dickinson and Company), and incubated with 5 µL of Annexin V FITC for 15 min at room temperature in the dark. Then, cells were stained with PI (final concentration: 20 μg/mL) straightway before measurement. Unstained and single-stained cells were used to determine the negative and positive boundaries and measured in each experiment. Annexin V^−^/PI^−^ cells were considered to be viable cells, Annexin V^+^/PI^−^ cells were considered to be early apoptotic cells, and Annexin V^+^/PI^+^ cells were considered to be late apoptotic/necrotic cells. Flow cytometry measurement was performed on FACSVerse™ (Becton, Dickinson and Company) and all data were analyzed by BD FlowJo™ software (Becton, Dickinson and Company).

### 2.6. Evaluation of Combined Inhibitor Application

The interaction among the inhibitors was evaluated by the Bliss independent model. The interaction of the inhibitor combination was determined by the difference between the observed (*E_O_*) and predicted (*E_P_*) inhibition of the combination therapy.

In double inhibitor application, *E_P_* was calculated with the following equation:*E_P_* = *E_A_* + *E_B_* − *E_A_* × *E_B_*,
where *E_A_* and *E_B_* are the relative inhibition of single-inhibitors *A* and *B*.

In triple inhibitor application, 
$E_{P}$
 was calculated with the following equation:*E_P_* = *E_A_* + *E_B_* + *E_C_* − *E_A_* × *E_B_* − *E_A_* × *E_C_* − *E_B_* × *E_C_* − *E_A_* × *E_B_* × *E_C_*,
where *E_A_*, *E_B_,* and *E_C_* are the relative inhibition of single-inhibitors *A*, *B*, and *C*.

*E_O_* > *E_P_* indicated a synergistic effect, *E_O_* = *E_P_* indicated an additional effect; *E_O_* < *E_P_* indicated an antagonistic effect. Bliss values for inhibitor combinations were calculated based on the results of proliferation and cell biomass of PDAC cell lines [28].

### 2.7. Examination of Cell Morphology Changes

Examination of PDAC cell line morphology changes was carried out by Pappenheim staining. After 72 h of exposure to the vehicle control, single inhibitor, or combined inhibitor, supernatants were collected and cells were harvested. After counting the cells, we resuspended the cell pellet and adjusted the cell density of the control group and each experimental group to 5 × 10^4^ cells/200 µL. Then, 200 µL of the cell suspension was fixed on a glass slide using Shandon Cytospin 3 centrifuge (Shandon, Frankfurt/Main, Germany), and two cell slides were made for each group. After 24 h of air-drying, the slides were stained with May–Grünwald solution (Merck, Darmstadt, Germany) for 6 min, washed with phosphate buffer solution (pH = 7.2) (Merck) three times for 1 min, then stained with Giemsa solution (1:10) (Merck) for 20 min, and washed with phosphate buffer solution three times for 1 min again. After the slides were air-dried for 24 h, the morphology of cells was examined and visualized with Evos XL Core Imaging System (Life Technologies, Darmstadt, Germany), magnified 100 times. Each experiment was repeated 3 times to eliminate random errors.

### 2.8. RNA Extraction

Total RNAs were extracted using the miRNeasy Mini Kit (QIAGEN GmbH, Hilden, Germany) according to the manufacturer’s instructions. For each cell line, only the RNA of the DMSO control group and the triple inhibitor application group were extracted. In brief, at least 5 × 10^6^ cells were harvested and washed twice with cold sterile PBS. Cell pellets were resuspended in 700 μL QIAzol Lysis Reagent (QIAGEN GmbH), then the aqueous phase that contains the total RNA of the lysed cells was extracted and purified by a silica membrane of RNeasy Mini spin columns. At last, total RNA was eluted by 30 μL of RNAse-free water.

After extraction, RNA concentrations, as well as OD 260/280 and OD 260/230 ratios, were measured with the NanoDrop 1000 Spectrophotometer (Thermo Fisher Scientific Inc., Waltham, MA, USA).

### 2.9. RNA Sequencing Analysis

The RNA quality was assessed using the Agilent RNA 6000 Nano Kit (Agilent Technologies Inc., Waldbronn, Germany) on the 2100 Bioanalyzer system (Agilent Technologies Inc.). Only samples with an RNA integrity number (RIN) >8 were proceeded to DNA library preparation using the Illumina Stranded mRNA Sample Preparation Kit (Illumina Inc., San Diego, CA, USA). Briefly, 800 ng of total RNA was enriched for mRNA via poly-T oligo-coated magnetic beads, and chemically fragmented under elevated temperature. The RNA fragments were then reverse-transcribed into the first- and second-strand cDNA using random hexamers. Double-stranded cDNA fragments were ligated with anchor primers and PCR-amplified for 10 rounds, using 10bp unique dual index primers (UDIs). The quality of the libraries was evaluated for fragment length distribution on the Agilent DNA-1000 Chip (Agilent Technologies Inc.). The library concentration was quantified using a Qubit dsDNA HS Assay kit (Life Technologies), normalized to 2 nM and equally pooled. The multiplexing library pool was sequenced for 2 × 101 bp paired-end reads at a final loading concentration of 750 pM on the NextSeq 2000 system and P3 Flow Cell at the sequencing facility of Research Institute for Farm Animal Biology (FBN), Dummerstorf, Germany.

### 2.10. Data Pre-Processing and Differentially Expressed Genes (DEGs) Analysis

Sample de-multiplexing and FASTQ generation of raw sequencing reads were conducted using on-board DRAGEN BCL Convert analysis workflow (Illumina). The data were quality-checked pre- and post-processing using FastQC version 0.11.9 [29]. Data pre-processing was performed using Trim Galore v.0.6.7 with the following options: -q 20—paired—stringency 3—length 20—illumine [29]. The remaining high quality paired reads were then aligned to the reference genome, Homo_sapiens.GRCh38 from Ensembl release 106 using Hisat2 version 2.2.1 [30]. The number of reads uniquely mapped to each gene was extracted from the HISAT2 mapping results using HTSeq version 2.0.1, with the following options: -f bam -r name—stranded = reverse -t exon -i gene_id -m union [31]. The resulting gene count data were further analyzed for DEGs using DESeq2 package [32]. DEGs that passed a threshold of|Log_2_(Fold Change)| > 1 and adjusted *p* value (padj) < 0.05 were considered analytically valuable and proceeded to Gene Ontology (GO) and Kyoto Encyclopedia of Genes and Genomes (KEGG) enrichment analysis.

The GO and KEGG enrichment analysis were applied for the functional annotation and pathway analysis using the gene set enrichment analysis (GSEA) [33,34]. The functional enrichment analyses of DEGs were explored by R package clusterProfiler4.0 and Pathview [35,36]. GO and KEGG enrichment analysis with a *p*-value < 0.05 and q-value < 0.25 were considered to have a significant impact and were selected for further analysis.

### 2.11. Statistical Analyses

Each experiment was performed in at least 3 biologically independent repetitions. Results of proliferation, biomass quantification, and apoptosis/necrosis analysis were expressed as mean ± standard deviation (SD). Statistical significance was determined by one-way ANOVA (after proving the data within each group conformed to the Gaussian distribution) or Kruskal–Wallis test (the data within each group conformed to the non-Gaussian distribution) and displayed as follows: *: *p* < 0.033, **: *p* < 0.002, ***: *p* < 0.001 versus the control group.

## 3. Results

### 3.1. KRAS Status of the PDAC Cell Lines

The analyzed seven PDAC cell lines were characterized by the following *KRAS* mutational statuses: one *KRAS* wild-type cell line (BXPC-3), one *KRAS* G12C (c.34G>T) cell line (MIA PACA-2), one *KRAS* Q61H (c.183A>C) cell line (T3M4), two *KRAS* G12D (c.35G>A) cell lines (ASPC-1and COLO357), and two *KRAS* G12V (c.35G>T) cell lines (CAPAN-1 and PATU8902). The information about each cell line includes the chromosomal location (#Chr), the zygosity (hom: homozygous, het: heterozygous), reference base (Ref), observed base (Obs), allele frequency (VAF), base change, and amino acid substitution, which are listed in Table 2. Thereby, COLO357 and T3M4 represent the only two cell lines characterized by a heterozygotic *KRAS* genotype.

### 3.2. Single Application of KRAS Inhibitors BI-3406 and Sotorasib to PDAC Cell Lines

The *KRAS* G12C inhibitor sotorasib had almost no inhibitory effect on the *KRAS* Q61H cell line T3M4 (Figure 1). At the highest tested concentration of 10 μM, cell proliferation and biomass were reduced by only 6% and 0%, respectively. In addition, sotorasib displayed similar inhibitory effects on *KRAS* wild-type and *KRAS* G12V cell lines, and the biomass of cell proliferation decreases ranged from 25% to 38% at the concentration of 10 μM. Notably, the inhibitory effects of sotorasib on ASPC-1 (VAF: 100) and COLO357 (VAF: 23.8), which both carry *KRAS* G12D, are quite different; cell proliferation decreased by 50% and 37%, and biomass decreased by 41% and 27%, respectively. Sotorasib appears to be more effective against *KRAS* G12D mutations with high VAF.

As expected, sotorasib showed a very strong inhibitory effect on MIA PACA-2, which carry a *KRAS* G12C mutation. A significant inhibitory effect can be observed at a concentration of 0.005 μM, while at 0.05 μM, cell proliferation and biomass were reduced by 69% and 60%, respectively (Figure 1 and Appendix A).

Compared with the DMSO control group, the KRAS::SOS1 inhibitor BI-3406 demonstrated a weak inhibitory effect on PDAC cell lines carrying KRAS G12V (CAPAN-1 and PATU8902). At the highest test concentration of 10 μM, cell proliferation only decreased by 11% and 17%, and the biomass decreased by 12% and 21%, respectively (Appendix A). In addition, the inhibitory effect of BI-3406 on the cell proliferation and biomass of the KRAS wild-type cell line BXPC-3 is similar to the inhibition observed in the KRAS G12V cell lines. The cell proliferation and biomass of BXPC-3 were reduced by only 15% and 27% at the concentration of 10 μM. BI-3406 demonstrated an increased, but still limited, inhibitory effect on the cell lines carrying the other three KRAS mutations (ASPC-1 and COLO357, KRAS G12D; MIA PACA-2, *KRAS* G12C; T3M4, *KRAS* Q61H). At the highest tested concentration, cell proliferation and biomass were only reduced between 30 and 50% (Figure 2 and Appendix A).

### 3.3. Combined Applications of KRAS, PI3K, and MEK1/2 Inhibitors Enhance Inhibition of PDAC Cell Lines

For BI-3406 in combination with trametinib and buparlisib, a significant increase in the inhibition of cell proliferation and biomass was observed when compared with the DMSO control group, regardless of whether double-inhibitor combinations or triple-inhibitor combinations were tested (Figure 3 and Appendix A). When comparing the effect of the triple-inhibitor with the effects of the double-inhibitor, a significantly increased inhibition in cell proliferation can also be observed in ASPC-1, BXPC-3, and MIA PACA-2. In CAPAN-1, a significant increase was only observed when comparing the triple therapy with the combination of BI-3406 and buparlisib. As for the other two combinations (BI-3406 + trametinib, trametinib + buparlisib), no significant increase could be observed. Moreover, we also observed similar inhibitory effects in the biomass quantification assay.

For the combination of sotorasib with trametinib and buparlisib, significant inhibition of cell proliferation and biomass was observed in the triple combination compared to the DMSO control group (Figure 4 and Appendix A). The addition of sotorasib significantly improved inhibition in ASPC-1 and MIA PACA-2 compared with a single application of trametinib or buparlisib. In addition, when focusing on the efficacy of the triple combination (sotorasib + trametinib + buparlisib) versus the double combination (trametinib + buparlisib), a significant increase in inhibitory effect was only observed in ASPC-1 and MIA PACA-2.

### 3.4. Bliss Analysis Revealed the Synergistic Effects of Double- and Triple-Application

The Bliss prediction effects were calculated based on the results of proliferation and biomass inhibition. For the BI-3406-based triple inhibitor combination, the Bliss predicted that inhibition (*E_P_*) is lower than the observed inhibition results (*E_O_*) in all cell lines (Figure 3). When focusing on comparing the double combination of trametinib + buparlisib and the triple combination of BI-3406 + trametinib + buparlisib, ASPC-1, BXPC-3, and MIA PACA-2 showed significantly higher inhibitory efficacy. However, this significant improvement did not appear in CAPAN-1, suggesting that BI-3406 was not able to enhance the inhibitory efficacy of trametinib + buparlisib in CAPAN-1. For the sotorasib-based triple inhibitor combination, *E_P_* was observed to be lower than *E_O_* in all cell lines. When focusing on comparing the double combination of trametinib + buparlisib and the triple combination of sotorasib + trametinib + buparlisib, only MIA PACA-2 demonstrated a significant improvement in inhibitory efficacy. Furthermore, in the other three cell lines, the inhibitory effects were not affected by the addition of sotorasib. These data indicated that the sotorasib-based triple inhibitor combination is synergistic in MIA PACA-2 cells that express the KRAS G12C mutant (using 0.005 µM sotorasib).

In the BI-3406-based double inhibitor combination, the combination of BI-3406 + trametinib demonstrated a significantly increased inhibitory effect in all four cell lines (Figure 3). The differences from *E_O_* and *E_P_* were between 20 and 43% (proliferation) and 23 and 36% (biomass) (Appendix A). The combination of BI-3406 + buparlisib also demonstrated a synergistic effect in all four cell lines; the differences between *E_O_* and *E_P_* were between 3 and 15% (proliferation) and 4 and 16% (biomass) (Appendix A). In addition, for sotorasib, either in combination with trametinib or in combination with buparlisib, synergistic effects were only observed in MIA PACA-2, with differences between *E_O_* and *E_P_* of 7%, 21% (proliferation) and 8%, 32% (biomass), respectively (Appendix A). In the other cell lines that do not harbor the KRAS G12C variant, the difference between *E_P_* and *E_O_* was almost 0, suggesting that sotorasib does not act synergistically in these cell lines when combined with trametinib or buparlisib.

### 3.5. Combined Application of KRAS, PI3K, and MEK1/2 Inhibitors Induce Apoptosis and Necrosis of PDAC Cell Lines

Apoptosis/necrosis assays were performed on ASPC-1, BXPC-3, CAPAN-1, and MIA PACA-2 cells after exposure to BI-3406-based inhibitor combinations and MIA PACA-2 after exposure to sotorasib-based inhibitor combinations. Compared to the DMSO control group, Annexin V/propidium iodide (PI) double staining revealed a significant increase in induced apoptosis/necrosis, when using the triple-inhibitor combinations (Appendix A). These triple-inhibitor combinations also significantly increased cell death when compared with all double-inhibitor combinations. In addition, most of the double-inhibitor combinations caused a significant increase in cell death when compared to the control group. Only in MIA PACA-2 cells, the combination of BI-3406 and buparlisib was not able to significantly increase cell death.

Furthermore, the microscopic evaluation at 100× magnification of Pappenheim stained samples indicated that the cells clearly demonstrated signs of cell death, including numerous vacuoles in the cytoplasm, splitting or breaking up of the nucleoli (karyorrhexis), protrusions of the plasma membrane, and apoptotic bodies, as well as morphological deformation (Figure 5 and Appendix A). These morphological changes were also observed in the samples that have been exposed to the double inhibitor combinations. However, there was more evidence after the application of the triple inhibitor combination.

### 3.6. Comparative Analysis of Differentially Expressed Genes (DEGs) between BI-3406 Combination-Treated and Non-Exposed PDAC Cell Lines

Differential expression analysis revealed several genes that were differentially regulated in triple combination-treated cells, when compared with the DMSO control exposed cells. For the combination of BI-3406 with trametinib and buparlisib, 587 DEGs were identified in ASPC-1 cells, 423 DEGs in BXPC-3 cells, 1191 DEGs in CAPAN-1 cells, and 1259 DEGs were identified in MIA PACA-2 cells (Appendix A). Of these DEGs, only 12 DEGs were shared among all the tested PDAC cell lines (Figure 6a). In addition, in the top 25 up- and down-regulated genes identified in the 4 cell lines (Figure 6b), no gene was shared by all cell lines.

### 3.7. Comparative Analysis of DEG Changes Induced by BI-3406 Combination-Treated and Sotorasib Combination-Treated in MIA PACA-2 Cell Line

For the sotorasib triple combination, only MIA PACA-2 cells were analyzed. Compared to the DMSO control group, 928 DEGs were identified in MIA PACA-2 (Appendix A). Comprehensive analysis of DEG changes in MIA PACA-2 using BI-3406 or sotorasib triple therapy revealed 778 DEGs that were up- or down-regulated by both inhibitor combinations (Figure 7a). In the top 25 up- and down-regulated genes, 17 overlapping up-regulated genes and 15 overlapping down-regulated genes were observed (Figure 7b).

### 3.8. Functional and Pathway Enrichment Analysis of DEGs Induced by Combination-Treated PDAC Cell Lines

In order to assess the effect of inhibitor combinations on PDAC cell lines, GO and KEGG pathway analysis was performed on all of the DEGs selected in result 3.6 for each cell line.

For the BI-3406 triple inhibitor combination, GO and KEGG enrichment analysis demonstrated different results in different PDAC cell lines. The number of GO terms, including the biological process (BP), the cellular component (CC), and the molecular function (MF), as well as the number of KEGG pathways caused by BI-3406 triple inhibitor combination treatment, are displayed in Table 3. Detailed information is displayed in Appendix A.

Further analysis of the GO term function revealed that in the PDAC cell lines, DEGs were involved in regulating the immune system, cell adhesion, cell migration, localization, locomotion, and response to stimulus in biological process, cell membrane, and extracellular functions in cellular components, as well as cytokine binding in molecular functions. KEGG pathway enrichment analysis identified nine overlapping pathways, which were involved in cancer, cellular community, cardiovascular disease, and immune regulation and directly acting on PI3K/AKT and TNF pathways (Appendix A). Furthermore, the expected RAS signaling pathway was not observed to be affected in all of the tested cell lines. The KEGG pathway results revealed that the RAS pathway was affected in ASPC-1, BXPC-3, and MIA PACA-2, but not in CAPAN-1.

For the sotorasib combination, 928 DEGs in MIA PACA-2 were involved in 849 BP, 39 CC, and 63 MF; KEGG analysis revealed that DEGs were enriched in 58 pathways, which are mainly associated with cancer, signal transduction, and the immune system (Appendix A).

Comparing the GO terms and KEGG pathway enrichment analysis of MIA PACA-2 in the two inhibitor combinations did not reveal major differences. The GO term demonstrated that both inhibitor combinations were involved in similar cellular functions in MIA PACA-2. KEGG analysis revealed that both inhibitor combinations were involved in immune regulation, signal transduction (especially PI3K/AKT, TNF, and JAK-STAT signaling pathways), metabolic activity, and cancer pathways (especially proteoglycans in cancer). The BI-3406 triple combination additionally participated in the MAPK pathway; however, this effect was not observed in the sotorasib triple combination (Appendix A).

## 4. Discussion

*KRAS* mutations are the most common mutations in PDAC patients and are characterized by poor prognosis and resistance to general treatment [6,14]. Although a series of targeted inhibitors have been developed for PDAC, so far, these inhibitors are still not routinely used in clinical treatment. In our study, sotorasib, which targets the *KRAS* G12C mutation, exhibited the expected inhibitory efficacy in MIA PACA-2, and significantly inhibited cell proliferation and biomass even at very low concentrations (0.005 μM). At the same time, sotorasib at 10 μM exhibited a partial inhibitory effect on other tested PDAC cell lines, except for T3M4 (KRAS Q61H). The cell proliferation and biomass decreased by 32–50% and 24–41%, respectively. However, in T3M4, minimal inhibition of cell proliferation and biomass was observed at all the concentrations tested. This may be due to the fact that the Q61H mutation has the lowest intrinsic GTPase activity and requires less upstream signaling to maintain a GTP-bound status [37]. In a previous report, the maximum plasma concentration of sotorasib was 7.5 μg/mL (13.4 μM) [38]. The results of this study demonstrated that sotorasib had inhibitory effects on KRAS G12D, G12V, and wild-type PDAC cell lines at a concentration of 10 μM, which can be achieved in clinical trials [38]. The incidence of serious adverse reactions at this concentration in clinical trials is low, suggesting that sotorasib can potentially become an interesting option for the development of novel approaches for the above-mentioned PDAC types [18,38]. Although sotorasib is currently only approved for the treatment of KRAS G12C-mutated NSCLC, the CodeBreaK100 study has confirmed its potential for the treatment of advanced KRAS G12C mutated PDAC with low side effects. At the same time, the clinical trials demonstrated that the maximum plasma concentration is higher than 10 μM [18]. Combined with our findings, sotorasib may also have inhibitory effects on KRAS G12D and G12V mutated PDAC in vivo, suggesting that sotorasib may have further potential to treat KRAS wild type and other KRAS G12-mutated PDACs besides KRAS G12C.

Using the multi-KRAS mutation inhibitor BI-3406, our results were comparable to those previously reported in 2D cultures [26]. The biological response of cell lines carrying the KRAS G12V mutation (CAPAN-1 and PATU8902) was similar to that of the wild-type cell line BXPC-3, showing a decrease of only about 15% at 10 μM. In the cell lines carrying KRAS G12C and G12D mutations, the inhibition of cell proliferation and biomass at 10 μM concentration was higher than 30%, up to 48.18%. In the previously reported in vivo studies, the BI-3406 single-inhibitor demonstrated a good inhibitory effect on KRAS G12C-mutated MIA PACA-2 cells, and the tumor volume of the two different doses of the experimental group was significantly reduced compared with the control group. However, even at the highest dose, BI-3406 was only able to inhibit tumor growth, but could not reduce tumor volume below the baseline [26]. It is suggested that a single application of BI-3406 does not have a strong inhibitory effect on PDAC cell lines both in vivo and in vitro. Nonetheless, it demonstrated a distinct synergistic effect with downstream pathway inhibitors in combination inhibition, especially with the MEK1/2 inhibitor trametinib. The combination of BI-3406 and trametinib demonstrated a synergistic effect in both *KRAS*-mutated and wild-type PDAC cell lines, which is in agreement with previous reports, both confirming the synergistic effect of BI-3406 and trametinib [26]. This is probably because BI-3406 combined with trametinib can block the negative feedback regulatory mechanism by reducing phosphorylated (p)-MEK and p-ERK levels, resulting in a strong synergistic effect [26]. Since this regulatory mechanism exists both in *KRAS*-mutated and wild-type cell lines, this double-inhibitor combination was also effective in the BXPC-3 cells. For the combination of BI-3406 and buparlisib, a synergistic effect was only observed in MIA PACA-2. Since buparlisib does not reduce p-MEK and p-ERK levels, it is highly likely that it fails to activate the negative feedback loop, resulting in a small synergistic effect [26].

The double-inhibitor combination based on sotorasib also displayed a synergistic effect, but mainly in MIA PACA-2 cells. Since RAS directly forms a complex with PI3K to further activate the PI3K signaling pathway, inhibition of these two proteins greatly reduces the activation of this pathway and might explain the synergistic effect of these two inhibitors [39,40,41,42]. Additive effects were observed in ASPC-1, BXPC-3, and CAPAN-1 cells, indicating that sotorasib might target an unknown target protein at a high concentration and the inhibition of this target protein does not synergistically interact with inhibitors of MEK and PI3K.

The efficacy of the triple-inhibitor combination of BI-3406, trametinib, and buparlisib was significantly stronger than that of the double-inhibitor combination in ASPC-1, BXPC-3, and MIA PACA-2 cells. However, in CAPAN-1 cells, there was no significant improvement in the triple-inhibitor combination versus the double-inhibitor combination of buparlisib and trametinib. Moreover, the KEGG pathway enrichment analysis revealed that in CAPAN-1, the RAS pathway was not affected by the triple therapy, while the enrichment of DEGs in the RAS pathway was observed in the other three cell lines. In addition, a single application of BI-3406 did not significantly inhibit the proliferation and biomass of CAPAN-1. Although BI-3406 has previously been reported to achieve good inhibitory effects on KRAS G12V-mutated NSCLC cell lines, this antitumor effect appears to be poor for PDAC cell lines [26]. This suggests that in PDAC cell lines, BI-3406 is less able to block the interaction between KRAS G12V and SOS1, at least not causing changes at the gene expression level. This may account for the low response of the KRAS G12V cell lines to BI-3406 and the inability of the BI-3406 to enhance the efficacy of downstream inhibitors in CAPAN-1. Moreover, the triple inhibitor combination of sotorasib demonstrated only an additive effect in ASPC-1, BXPC-3 and CAPAN-1, further confirming that the inhibition of non-KRAS G12C mutant cell lines by sotorasib is not affected by the changes in the RAS/RAF/MEK/ERK pathway or PI3K/AKT pathway.

The BI-3406 triple inhibitor combination modulated immunity, cell adhesion, migration, and targeted cancer pathways in all four cell lines. This indicates that this inhibitor combination can directly influence the pathophysiology of tumor cells, but might also indirectly inhibit the growth of PDAC cells by modulating the immune system, as well as cell-to-cell interactions. Furthermore, we observed that in all four cell lines, both triple inhibitor combinations regulated DEGs, which are involved in the response to hypoxia. These genes (*ALDOA*, *IL6*, *IL6R*, *EGF*, *VEFG*, *PDK-1*, *ENO1*, etc.) were all associated with the hypoxia inducible factor-1 (HIF-1) pathway, suggesting that both combinations can act on the HIF-1 pathway. Several studies have shown that HIF-1 is associated with tumor growth in a variety of cancers, including PDAC [43]. Inhibition of mTOR blocks the translation of HIF-1 mRNA, and inhibition of ERK can also lead to inhibition of HIF-1 [44,45]. The combination of the two inhibitors in this experiment affected both mTOR and ERK, leading to changes in the downstream HIF-1 pathway, which seems to be another mechanism of this inhibitor combination.

Altogether, our current study demonstrates the antiproliferative effects of KRAS inhibitors alone or in combination with downstream inhibitors in PDAC cell lines in vitro. Moreover, the dose of each inhibitor was greatly reduced when used in combination, thereby reducing the side effects of the inhibitor. The KRAS::SOS1 inhibitor BI-3406 was able to enhance the antiproliferative effect of downstream inhibitors in the KRAS wild-type, KRAS G12C, and KRAS G12D mutant cell lines, but not for the KRAS G12V mutant cell lines. The KRAS G12C inhibitor sotorasib mainly enhanced the anti-proliferative effect of downstream inhibitors in KRAS G12C mutant cell lines.

## 5. Conclusions

Our current study demonstrates the effects of two KRAS inhibitors, BI-3406 and sotorasib, as monotherapy for PDAC. This provides evidence for a potential extended application of sotorasib in non-KRAS G12C mutated PDAC and the application of BI-3406 as a multi-KRAS mutated inhibitor in PDAC. In addition, these two KRAS inhibitors act synergistically or additively with downstream pathway inhibitors, when reducing cell proliferation and biomass in PDAC cell lines with different *KRAS* statuses. The two triple combinations also demonstrated extraordinary effects in enhancing inhibitor efficacy and reducing inhibitor dose. These data emphasize the importance of KRAS as a therapeutic target for PDAC and validate two different mechanisms of KRAS inhibition and its interaction with downstream pathway inhibitors. The current study provides novel ideas for the drug treatment of PDAC; however, in vivo experiments and clinical trials are still needed to observe the real efficacy and adverse reactions of these inhibitors and inhibitor combinations for the treatment of PDAC.

## Figures and Tables

**Figure 1 cancers-14-04467-f001:**
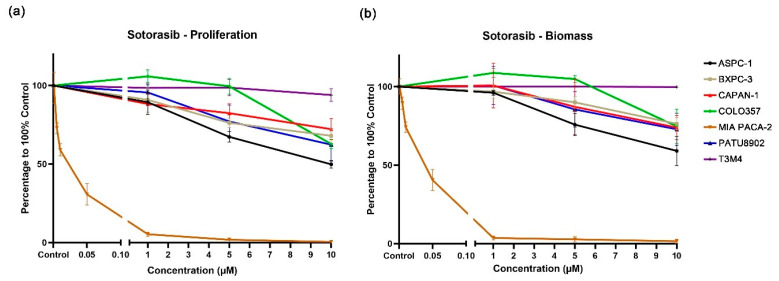
Proliferation (**a**) and biomass (**b**) changes in PDAC cell lines after exposure to different concentrations of sotorasib.

**Figure 2 cancers-14-04467-f002:**
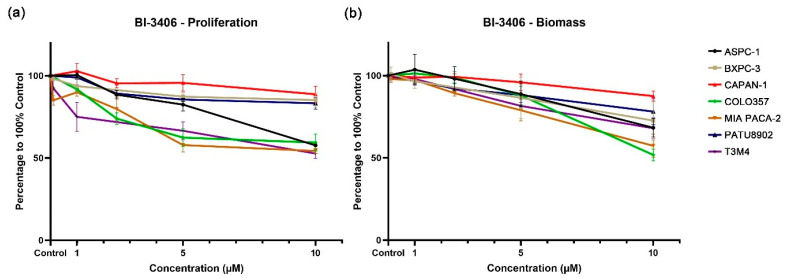
Proliferation (**a**) and biomass (**b**) changes in PDAC cell lines after exposure to different concentrations of BI-3406.

**Figure 3 cancers-14-04467-f003:**
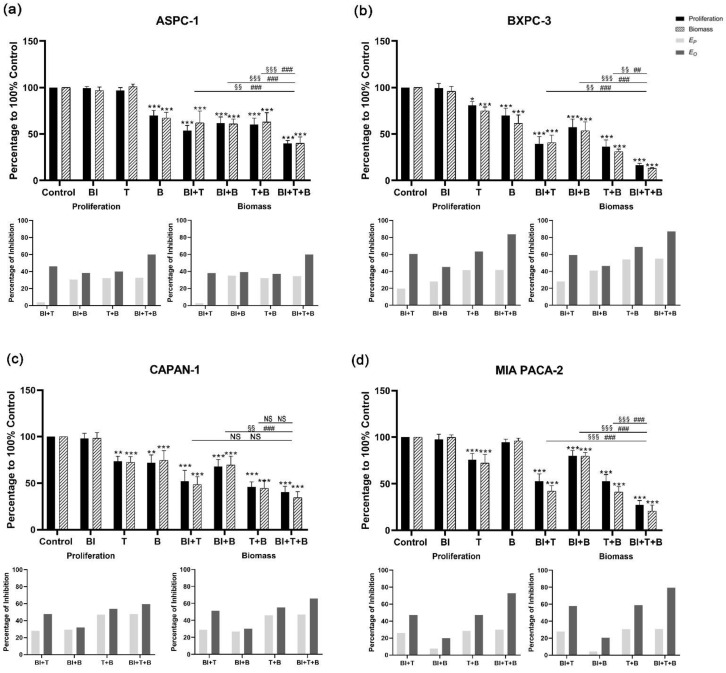
Cell proliferation and biomass of ASPC-1 (**a**), BXPC-3 (**b**), CAPAN-1 (**c**), and MIA PACA-2 (**d**) after 72 h BI-3406, trametinib, buparlisib or inhibitor combination exposure, as well as analysis of synergistic effect using Bliss independent model. Data are presented as mean ± SD. Significance of a treatment effect compared to the DMSO control was determined by one-way ANOVA and displayed as *: *p* < 0.033, **: *p* < 0.002, ***: *p* < 0.001 (*n* ≥ 3). The significance of the treatment effect for double inhibition compared to triple inhibition was determined by one-way ANOVA and was shown as # (proliferation), § (biomass): *p* < 0.033; ##, §§: *p* < 0.002, ###; §§§: *p* < 0.001. BI: BI-3406; T: trametinib; B: buparlisib; NS: not significant; *E_P_*: predicted inhibition by Bliss independent model; *E_O_*: observed inhibition.

**Figure 4 cancers-14-04467-f004:**
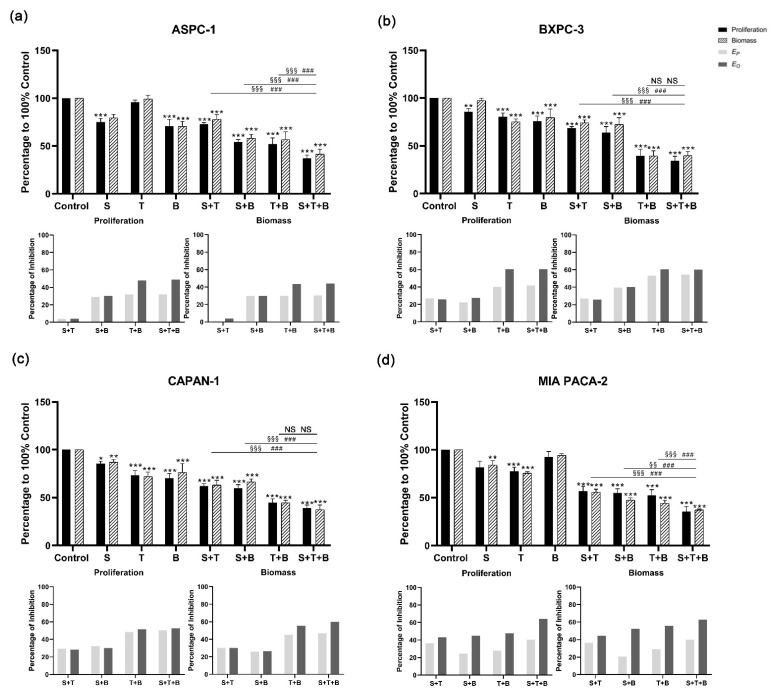
Cell proliferation and biomass of ASPC-1 (**a**), BXPC-3 (**b**), CAPAN-1 (**c**), and MIA PACA-2 (**d**) after 72 h sotorasib, trametinib, buparlisib or inhibitor combination exposure, as well as analysis of synergistic effect using Bliss independent model. Data are presented as mean ± SD. Significance of a treatment effect compared to the DMSO control was determined by one-way ANOVA and displayed as *: *p* < 0.033, **: *p* < 0.002, ***: *p* < 0.001 (*n* ≥ 3). The significance of the treatment effect for double inhibition compared to triple inhibition was determined by one-way ANOVA and was shown as # (proliferation), § (biomass): *p* < 0.033; §§: *p* < 0.002, ###; §§§: *p* < 0.001. S: sotorasib; T: trametinib; B: buparlisib; NS: not significant; *E_P_*: predicted inhibition by Bliss independent model; *E_O_*: observed inhibition.

**Figure 5 cancers-14-04467-f005:**
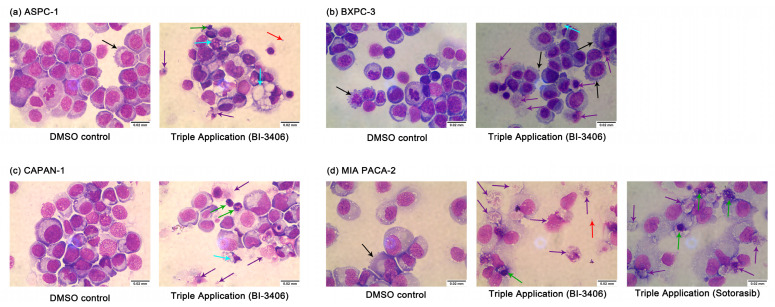
Morphology changes in ASPC-1 (**a**), BXPC-3 (**b**), CAPAN-1 (**c**), and MIA PACA-2 (**d**) after DMSO or triple inhibitor combination exposure. Magnification: 100×. ↑ membrane bubbles, membrane bound apoptotic body, ↑ vacuolization, ↑ apoptotic body, ↑ nuclear condensation/fragmentation; ↑ rupture of the plasma membrane.

**Figure 6 cancers-14-04467-f006:**
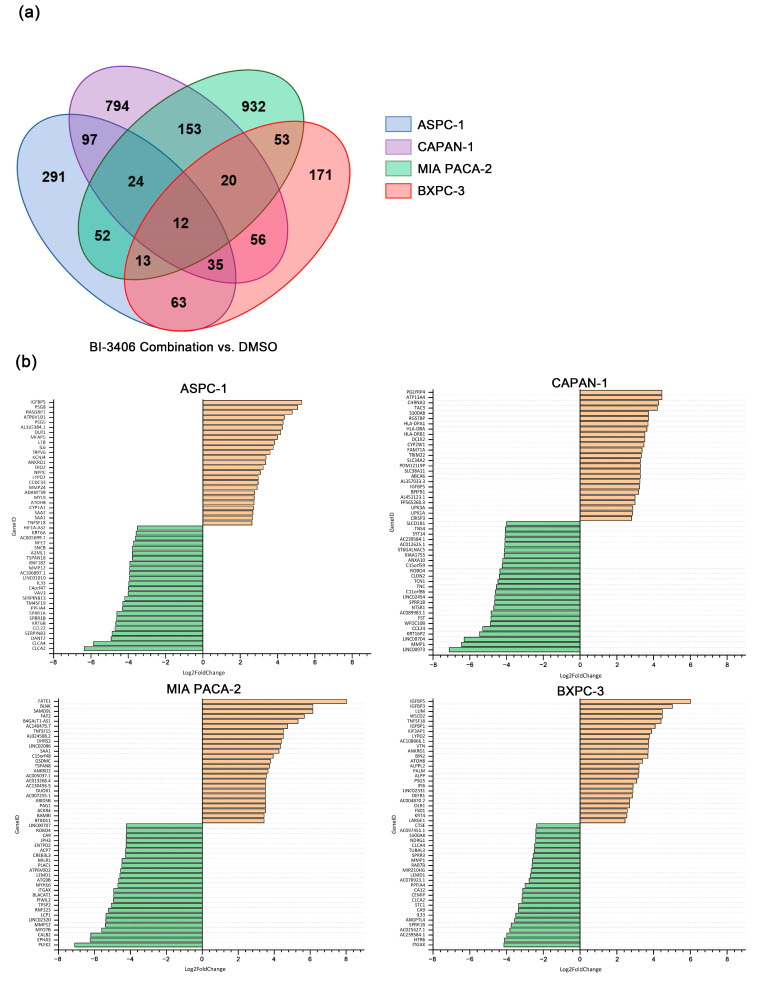
Number and overlap of DEGs in ASPC-1, BXPC-3, CAPAN-1 and MIA PACA-2 cell lines after exposure to BI-3406 combination (**a**) and the top 25 up- and down-regulated genes before and after BI-3406 combination exposure (**b**).

**Figure 7 cancers-14-04467-f007:**
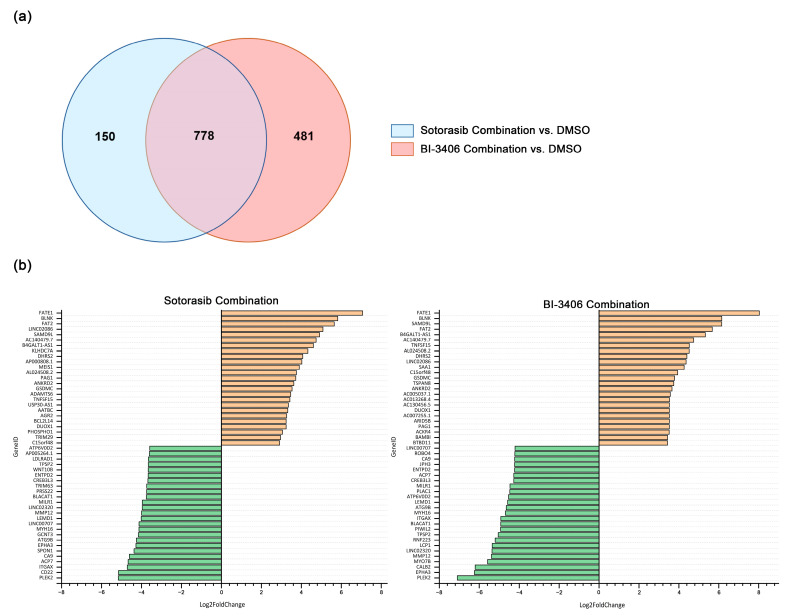
Number and overlap of DEGs in MIA PACA-2 cell lines after exposure to sotorasib combination or BI-3406 combination (**a**) and the top 25 up- and down-regulated genes before and after inhibitor combination exposure (**b**).

**Table 1 cancers-14-04467-t001:** Inhibitor concentrations used for combined application.

Cell Lines	BI-3406	Sotorasib	Trametinib	Buparlisib
ASPC-1	4 μM	4 μM	0.001 μM	0.3 μM
BXPC-3	4 μM	4 μM	0.001 μM	1 μM
CAPAN-1	4 μM	4 μM	0.005 μM	0.3 μM
MIA PACA-2	4 μM	0.005 μM	0.0025 μM	0.6 μM

**Table 2 cancers-14-04467-t002:** *KRAS* status of PDAC cell lines.

Cell Line	#Chr	Start	End	Ref	Obs	Zygosities	VAF	Gene	Base Change	AA Change
BXPC-3	chr12	25398284	25398284	G	G	hom	100	*KRAS*	-	-
ASPC-1	chr12	25398284	25398284	G	A	hom	100	*KRAS*	NM_033360.2:c.35G>A	G12D
COLO357	chr12	25398284	25398284	G	A	het	23.8	*KRAS*	NM_033360.2:c.35G>A	G12D
CAPAN-1	chr12	25398284	25398284	G	T	hom	97.1	*KRAS*	NM_033360.2:c.35G>T	G12V
PATU8902	chr12	25398284	25398284	G	T	hom	100	*KRAS*	NM_033360.2:c.35G>T	G12V
MIA PACA-2	chr12	25398285	25398285	G	T	hom	99.6	*KRAS*	NM_004985.5:c.34G>T	G12C
T3M4	chr12	25380275	25380275	A	C	het	32.6	*KRAS*	NM_033360.2:c.183A>C	Q61H

**Table 3 cancers-14-04467-t003:** GO term and KEGG pathway enrichment analysis of DEGs induced by BI-3406 triple inhibitor combination treatment.

PDAC Cell Line	KRAS Mutation	GO Term			KEGG Pathway
		Biological Process	Cellular Components	Molecular Functions	
BXPC-3	Wild Type	847	49	96	24
ASPC-1	KRAS G12D	744	80	96	48
CAPAN-1	KRAS G12V	1447	116	168	59
MIA PACA-2	KRAS G12C	1053	76	120	66

## Data Availability

The data supporting the reported results can be found on the website in detail in the article.

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
