# Peer review of "Inhibition of KRAS, MEK and PI3K Demonstrate Synergistic Anti-Tumor Effects in Pancreatic Ductal Adenocarcinoma Cell Lines"

_cancers, 2022, doi:10.3390/cancers14184467_

Round 1

Reviewer 1 Report

The authors did a thorough study on comparison of different KRAS inhibitors over a series of cell lines. This article discussed inhibition effects of these KRAS inhibitors including cell proliferation, cell apoptosis induction, cell morphology and synergistic effect when BI-3406 and sotorasib used in combination. Overall, the manuscript did a comprehensive investigation regarding the KRAS inhibitors will attract wide interest in the field of cancer therapy. Here are some comments to the authors.

1.       Could the authors provide more rationale/references regarding inhibitor treatment concentrations?  

2.       I would suggest the authors to perform the proliferation curve with more concentrations on the lower end as the proliferation rate dropped significantly within the first three data point. Thus, the curve could provide a more precise IC50 value for the inhibitors.

3.       Add scale bar in the imaging results.

4.       What’s the potential side effect of the KRAS inhibitors? How to avoid off-target cytotoxicity?

5.       Could the author include more discussion regarding in vivo performance of these KRAS inhibitors? 

Author Response

  1. Could the authors provide more rationale/references regarding inhibitor treatment concentrations?  

Response 1: We are very grateful for this comment. The concentrations were chosen based on our previous in vitro experiments and published literature. The revised sentences can be found in Method part Line 160-165.

“The results of single inhibitor application and related experiments were comprehensively analyzed, and specific PDAC cell lines and inhibitor concentrations were selected for further combined application experiments and the concentrations are listed in Table 1 (The results of the Buparlisib inhibition assay are detailed in a previously published literature, and the results of the Trametinib inhibition assay are detailed in the Supplementary Table S14)”

  1. I would suggest the authors to perform the proliferation curve with more concentrations on the lower end as the proliferation rate dropped significantly within the first three data point. Thus, the curve could provide a more precise IC50 value for the inhibitors.

Response 2: We greatly appreciate this comment. In our study design the performed inhibition assay of the single KRAS inhibitor had the purpose to identify a suitable inhibitor concentration for combined inhibition. Thus, the primary intention was not to calculate the corresponding inhibitor IC50 value. Therefore, more concentration gradient tests at low concentrations were not performed in this experiment. However, we appreciate the comment and will perform more concentrations in further experiments and refine the proliferation curve to obtain precise IC50 values.

  1. Add scale bar in the imaging results.

Response 3: We apologize for the missing of the scale bar. The scale bar has been added to the imaging results.

  1. What’s the potential side effect of the KRAS inhibitors? How to avoid off-target cytotoxicity?

Response 4: We thank the reviewer for the comment as potential side effects are a key aspect when introducing novel therapeutic approaches. Accordingly, we added recent result of clinical trials with Sotorasib into the manuscript. The revised section can be found in Introduction part Line 77-82:

“According to the recently disclosed CodeBreaK 100 clinical trial results, Sotorasib displayed good efficacy in the treatment of advanced KRAS G12C-mutated PDAC with 8 of the 38 patients having a partial response and 32 of 38 patients having disease control. The side effects of Sotorasib are described as mild as only a few patients were affected by grade 3 diarrhea, fatigue, and abdominal pain, no grade 4 side effects were observed in patients.”

For BI-3406, no clinical trials have been conducted, so there are no reports of related side effects. In animal experiments, mice treated with a dose of 50 mg/kg showed good tolerance. The revised sentence can be found in Introduction part Line 111-112:

“Moreover, the experimental animals displayed good tolerance to BI-3406 treatment.”

It has not yet been reported whether the two KRAS inhibitors have other targets. Therefore, at this point we cannot give a clear statement whether there is off-target toxicity at present. In addition, both KRAS inhibitors were tolerated in previous clinical trials and in vivo trials. Therefore, it is to assume that even if off-target toxicity occurs, this side effect can be controlled and will not cause serious damage to the patient.

  1. Could the author include more discussion regarding in vivo performance of these KRAS inhibitors? 

Response 5: We are very grateful for this comment and add more discussion regarding in vivo performance of the KRAS inhibitors. The revised sentences can be found in Discussion part Line 490-497 and Line 503-509

“Although Sotorasib is currently only approved for the treatment of KRAS G12C-mutated NSCLC, the CodeBreaK100 study has confirmed its potential for the treatment of advanced KRAS G12C mutated PDAC with low side effects. At the same time, the clinical trials demonstrated than the maximum plasma concentration is higher than 10 μM. Combined with our findings, Sotorasib may also have inhibitory effects on KRAS G12D and G12V mutated PDAC in vivo, suggesting that Sotorasib may have further potential to treat KRAS wild type and other KRAS G12-mutated PDACs besides KRAS G12C.”

“In the previously reported in vivo studies, BI-3406 single-inhibitor demonstrated a good inhibitory effect on KRAS G12C-mutated MIA PACA-2 cells, and the tumor volume of the two different doses of the experimental group was significantly reduced compared with the control group. However, even at the highest dose, BI-3406 was only able to inhibit tumor growth, but could not reduce tumor volume below baseline. It is suggested that BI-3406 single application does not have strong inhibitory effect on PDAC cell lines both in vivo and in vitro.”

Reviewer 2 Report

Ma et al assessed the antitumor efficacy of two KRAS inhibitors (namely BI-3406 and Sotorasib) alone or in combination with downstream signalling: MEK1/2 inhibitor (Trametinib) and PI3K inhibitor (Buparlisib) in different pancreatic ductal adenocarcinoma (PDAC) cell lines e.g., those with mutated KRAS (ASPC-1, CAPAN-1, BXPC-3, COLO357, PATU8902, and T3M4) or wild-type KRAS (BXPC-3). The study showed that both KRAS inhibitors, when used in combination with Trametinib and Buparlisib, their efficacies are significantly increased by a synergistic or additive effect at least in one cell lines. They also employed a state-of-art omic technology in an attempt to identify the molecular mechanisms/pathways underpinning this synergistic/additive effect of KRAS inhibitor and PI3K inhibitor combination. The study is properly designed, and the results are appropriately presented. The manuscript is well-written. I have a few minor comments.

Comments

1)      Please consider reading the manuscript carefully for minor typos correction and also references

2)      Why 72 hours of cell cultures was chosen for the analysis of cell viability, proliferation, morphology etc?

3)      Why only triple inhibitors treatment/groups were considered for total RNA isolation and sequencing?

4)      Please remove “figure 1, figure 2, and so on” on the top of each figure, this has been mentioned in the legend of each figures

5)      Figure 1, internal legend  (cell line names) is not clear in figure 1and try to reduce the y-axis scale e.g., to 100 or 105/110% and also hide the minor tick . Whats was the control used to calculate these effect, is it cells treated with the vehicle? Why are not shown?

6)      Figure 3 and 4, the authors compared single and double treatment to control but there is a lack of statistical comparison of double treatment to single treatment (e.g, BI+T, BI+B, S+T and S+B) vs BI or S), this is particularly important in case of Sotorasib as treatment with S alone has induced significant reduction in proliferation and/ biomass.

7)      Figure 5, please note that the legend is hided by the photos

8)      Figure 7 the authors may consider rearranging this figure in order to improve the quality/clarity of the figures as the name of genes up or downregulated on vertical axis are not clearly seen.

9)      Please consider improving the conclusion  to support the results and highlight the future prospective

Author Response

1) Please consider reading the manuscript carefully for minor typos correction and also references

Response 1: We appreciate this comment, the article has been checked by an English native speaker, typos and erroneous references have been corrected.

2) Why 72 hours of cell cultures was chosen for the analysis of cell viability, proliferation, morphology etc?

Response 2: We are grateful to this comment, in our previous experiments with PDAC cell lines, when the 48 hour time point was used, a sufficient number of cells could not be obtained for the experiment, and at 96 hours the cells proliferated too much to accurately calculate the actual number of cells in the control group. Therefore, 72 hours was chosen as the experimental time point for cell viability, proliferation, morphology, etc.

3) Why only triple inhibitors treatment/groups were considered for total RNA isolation and sequencing?

Response 3: We thank the reviewer for this question. For single-inhibitor and double-inhibitor application, previous studies have reported the efficacy and mechanisms, so our study focused on the antitumor effect and mechanism of triple-inhibitor application on PDAC cell lines [1,2]. Therefore, for single-inhibitor and double-inhibitor applications, we only observed the inhibition of cell proliferation and biomass, as well as the induction of apoptosis/necrosis, and compared with previous research results to comprehensively evaluate the effects of single-inhibitor and double-inhibitor applications. For the triple-inhibitors application, there were no corresponding studies for reference. Therefore, we considered isolating total RNA and sequencing for triple application to explore the mechanism of the two triple-inhibitors application.

[1] Hofmann, M.H.; Gmachl, M.; Ramharter, J.; Savarese, F.; Gerlach, D.; Marszalek, J.R.; Sanderson, M.P.; Kessler, D.; Trapani, F.; Arnhof, H.; et al. BI-3406, a Potent and Selective SOS1-KRAS Interaction Inhibitor, Is Effective in KRAS-Driven Cancers through Combined MEK Inhibition. Cancer Discov 2021, 11, 142-157, doi:10.1158/2159-8290.CD-20-0142.

[2] Brown WS, McDonald PC, Nemirovsky O, Awrey S, Chafe SC, Schaeffer DF, Li J, Renouf DJ, Stanger BZ, Dedhar S. Overcoming Adaptive Resistance to KRAS and MEK Inhibitors by Co-targeting mTORC1/2 Complexes in Pancreatic Cancer. Cell Rep Med. 2020 Nov 17;1(8):100131. doi: 10.1016/j.xcrm.2020.100131.

4) Please remove “figure 1, figure 2, and so on” on the top of each figure, this has been mentioned in the legend of each figures

Response 4: We are grateful to this comment, the correspondence content have been removed.

5) Figure 1, internal legend  (cell line names) is not clear in figure 1and try to reduce the y-axis scale e.g., to 100 or 105/110% and also hide the minor tick . Whats was the control used to calculate these effect, is it cells treated with the vehicle? Why are not shown?

Response 5: We really appreciate this suggestion, the legend has been adjusted for a sharper image, the y-axis scale has been reduced, and the minor ticks have been hidden. The controls used to calculate these effects are vehicle-treated cells, and the corresponding results are shown as control, at the zero point of the x-axis

6) Figure 3 and 4, the authors compared single and double treatment to control but there is a lack of statistical comparison of double treatment to single treatment (e.g, BI+T, BI+B, S+T and S+B) vs BI or S), this is particularly important in case of Sotorasib as treatment with S alone has induced significant reduction in proliferation and/ biomass.

Response 6: We apologize for the lack of statistical comparison of double treatment to single treatment. The statistical comparisons of double treatment versus monotherapy have been added to Supplementary Tables S3 and S5. Since our study focused on comparing triple treatment with double and single treatment, the statistical comparison of double treatment and single treatment is not displayed in the main text figure. In addition, after adding the statistical comparison of double treatment and single treatment, the content of the figure will be very confusing and difficult to find the focus. Therefore, the statistical comparisons for double treatment and single treatment are only displayed in the Supplementary Table S3 and S5.

7) Figure 5, please note that the legend is hided by the photos

Response 7: We appreciate this comment, the position of the photo and legend has been adjusted.

8) Figure 7 the authors may consider rearranging this figure in order to improve the quality/clarity of the figures as the name of genes up or downregulated on vertical axis are not clearly seen.

Response 8: We thank the reviewer for this comment. The figure 6 and figure 7 have been rearranged and the clarity has been enhanced.

9) Please consider improving the conclusion to support the results and highlight the future prospective

Response 9: we are grateful to this comment, the conclusion has been improved and revised sentences can be found in Conclusion part Line 582-584, 586-588 and 590-593:

“This provides evidence for a potential extended application of Sotorasib in non-KRAS G12C mutated PDAC and the application of BI-3406 as a multi-KRAS mutated inhibitor in PDAC.”

“The two triple combinations also demonstrated extraordinary effects in enhancing inhibitor efficacy and reducing inhibitor dose.”

“The current study provides novel ideas for the drug treatment of PDAC, however, in vivo experiments and clinical trials are still needed to observe the real efficacy and adverse reactions of these inhibitors and inhibitor combinations for the treatment of PDAC.”